# Potential Health Risk of Aluminum in Four *Camellia sinensis* Cultivars and Its Content as a Function of Leaf Position

**DOI:** 10.3390/ijerph191911952

**Published:** 2022-09-21

**Authors:** Huijuan Yang, Yan Chen, Jennifer M. Shido, Randall T. Hamasaki, Wayne T. Iwaoka, Stuart T. Nakamoto, Haiyan Wang, Qing X. Li

**Affiliations:** 1College of Tobacco Science, Henan Agricultural University, Zhengzhou 450002, China; 2Key Laboratory of Nuclear Agricultural Sciences of Ministry of Agriculture and Zhejiang Province, Institute of Nuclear Agricultural Sciences, Zhejiang University, Hangzhou 310058, China; 3Department of Human Nutrition, Food and Animal Sciences, University of Hawaii at Manoa, Honolulu, HI 96822, USA; 4Department of Plant and Environmental Protection Sciences, University of Hawaii at Manoa, Honolulu, HI 96822, USA; 5Department of Molecular Biosciences and Bioengineering, University of Hawaii at Manoa, Honolulu, HI 96822, USA

**Keywords:** aluminum, *Camellia sinensis*, leaf position, health risk, tea

## Abstract

Tea plants can accumulate aluminum (Al) in their leaves to a greater extent than most other edible plants. Few studies, however, address the Al concentration in leaves at different positions, which is important information for tea quality control. Leaves from four different cultivars of *Camellia sinensis* L. grown in Hawaii were analyzed for Al concentrations at 10 different leaf positions. Each cultivar was harvested in the winter and summer to determine seasonal variations of Al concentrations in the leaves. The results showed that Al concentrations in the winter leaves were an average of 1.2-fold higher than those in the summer leaves, although the seasonal variations were not statistically significant. The total Al concentration of successively lower leaves showed an exponential increase (R^2^ ≥ 0.900) for all four cultivars in the summer season, whereas those of the winter leaves fit a bi-phase linear regression (R^2^ ≥ 0.968). The regression of the Al concentrations against the top-5 leaf positions in the winter season fit one linear regression, while that against leaf positions 6–11 fit another linear regression. The average Al concentrations between the third leaf and the shoot plus first two leaves increased approximately 2.7-fold and 1.9-fold for all cultivars in the winter and summer months, respectively. The Al concentrations in the rest of the leaves increased approximately 1.5-fold in a sequential order. The target hazard quotient being between 1.69 × 10^−2^ and 5.06 × 10^−1^ in the tea leaf samples of the four cultivars in Hawaii were all less than 1, suggesting negligible health risks for consumers. The results of this study may be useful for directing harvest practices and estimating tea quality.

## 1. Introduction

Tea is the second most consumed beverage worldwide [1] and is becoming increasingly popular with an annual average consumption of 155 cups per person in the U.S. [2]. Annually, the U.S. imports approximately 107,414 tons of tea worth more than $474 million [3]. It is well known that the tea plant accumulates Al in its leaves to a greater extent than most other edible plants, where it is essential for tea root growth by maintaining DNA integrity in meristems [4]. Culturally, tea prefers soils with pH values ranging from 4.5 to 5.5 in well-watered tropical environments, such as those across parts of Hawaii. Since the daily allowable intake (DAI) of Al is 6–14 mg/day for teens and adults, as set by Joint FAO/WHO Expert Committee on Food Additives [5], it is appropriate to understand how Al concentrations in fresh tea leaves vary across countries and varieties and with season and position on the plant.

Information already exists on Al concentrations in tea leaves from different origins, such as China, India, Japan, Sri Lanka, Turkey, and Vietnam [6], but was lacking for Hawaii, where there is increasing interest in the cultivation and commercial exploitation of multiple cultivars of tea being evaluated in experimental plantations. Knowledge gaps of Al in tea include a lack of understanding on the effects of seasonal conditions and leaf positions on Al concentrations and how any differences might affect DAIs of Al in the end-product.

Much research has been done in the analysis of Al concentrations in tea products because of the underlying correlation between dietary Al intake and potential toxicities, for example, Alzheimer’s disease, osteodystrophy, and kidney failure [7,8]. Al concentrations found in commercial tea products ranged from 123 to 3260 µg/g [6,9]. Al can be transferred into tea infusions through brewing tea, then enter the human body via tea drinking, thereby causing potential harm to human health. In addition, Al is present in food, such as fruits, vegetables, and cereal products, and is used in various processed foods as a common food additive [10,11].

The mechanism of Al enrichment in tea trees is very complex, and is related to the age and organs of tea trees, soil physicochemical properties, agronomic practices, etc. [12]. In Hawaii, tea plants are grown in quite similar conditions [13]. Al concentrations increase with leaf age, with at least 10-fold greater concentrations in old leaves compared to young leaves [14]. The descriptors “young” and “old” leaves are often labeled as being found at the top and bottom of the bush, respectively, and/or according to leaf size or by estimated age of the tea plant [14,15]. “Young” leaves are often referred to as shoots/bud plus two or three leaves. However, the definition for “old” or “mature” leaves is unclear. For example, “old” may refer to the leaves found at the base of the branches [15] from a 10-year-old tea bush [16], or provide no definition at all. The typical concentrations of Al found in the “young” leaves ranged from 250 to 660 μg/g (dry weight), and higher levels of Al in “old” leaves ranged from 4300 to 10,400 μg/g (dry weight) [12]. Those studies focused on analyzing the localization of Al in the plants to understand the uptake and transport, and tolerance and resistance to stresses, with little research on factors such as leaf positions, seasons, and cultivars. In this study, we examined Al concentrations in leaves from different leaf positions of four *C. sinensis* cultivars grown in Hawaii, analyzed these for total Al; assessed the differences in Al among leaf positions, seasons, and cultivars; and then considered the implications for human health and tea quality.

## 2. Materials and Methods

### 2.1. Study Site

The study site was at the University of Hawaii, College of Tropical Agriculture and Human Resources, Mealani Research Station in Waimea, Hawaii. The tea production field at Mealani is approximately 610 m above sea level. Annual mean temperatures range from 13 °C to 21 °C. The soil is a Honokaa series in the Andisols order [17]. The mean annual rainfall of 165.1 cm was supplemented by timer-operated drip irrigation. The tea fields at Mealani were planted from 1999–2004. Soil and tissue tests were used to guide fertilization, typically by monthly application of granular fertilizers supplemented by fertigation as needed. The pH of the 0–15 cm layer soil typically ranges from 5.3 to 5.5 (1:1, soil/water).

### 2.2. Sample Collections

The plots were left unharvested to allow shoots to grow to desired lengths, and a minimum of ten stems with undamaged leaves were selected for each of three replicates per cultivar. Fresh leaf samples from four cultivars of *C. sinensis* (Yabukita, Yutaka Midori, Mealani, and Ohiwase) for 10 data points encompassing 11 leaf positions commencing at the stem tip (shoot plus two leaves, leaf 3…leaf 11) were collected for subsequent analysis for total Al. The cultivars were chosen because they are of interest for distribution to commercial tea growers in the State of Hawaii, USA. Representative leaf samples of each cultivar were processed separately by cultivar and replication. Specifically for each leaf position, leaves including petioles were individually detached from the stem, wiped with a clean damp cloth to eliminate surface contaminants, and then placed in a package labelled with the cultivar name, leaf position, and replicate number for shipment to the Agricultural Diagnostic Service Center, University of Hawaii at Manoa. Moreover, leaf samples (with petioles) from each cultivar were collected in triplicate during the summer (July 2009) and winter (February 2010) to determine seasonal variability. Leaf positions 7 and 9 were not part of the original sampling scheme since they were known to be older than leaves typically harvested. Those leaf positions were added in the subsequent winter-based sampling due to the observed differences between adjacent leaf positions (see Figure 1 and Figure 2). In addition, five different commercial, powdered tea samples were purchased in Hawaii for comparison with dry leaf and infusion data (Table 1). The powdered samples were labeled as “matcha”, which are products of Japan. All leaf and powdered tea samples were stored in moisture-proof containers at room temperature in the dark until analysis.

Shoot-plus-two-leaf samples of the Yutaka Midori cultivar were used for infusion preparation. These were prepared by steeping 2.0 g portions in 200 mL of deionized boiling water (18 mΩ) for 30 min in a covered container. The resulting infusions were filtered through Whatman no. 42 paper, then diluted to 250 mL in volumetric flasks and allowed to cool to room temperature prior to analysis. Infusion samples were prepared in triplicate and stored in covered plastic vials at 3–5 °C until analyzed. No precipitations occurred within the stored infusions.

### 2.3. Chemical Analysis

The samples of tea leaves were further cleaned with damp paper towels to remove visible dust, placed in a forced-draft oven at 55 °C for 12 h, then ground with a Wiley mill to pass 2 mm. A 0.50 g sample was dry-ashed in a porcelain crucible for 4–6 h at 500 °C in a muffle furnace, cooled, dissolved in 1 mol/L HNO_3_, evaporated to dryness, then ashed again for 1 h. The resulting residue was dissolved in 25 mL of 1 mol/L HCl [21]. All prepared samples (dried leaf, powdered tea, and infusion) were then analyzed for total Al by inductively-coupled plasma optical emission spectrometry (ICP-OES), using a wavelength of 396.152 nm according to the AOAC Official Method 953.01 [22,23]. The laboratory’s contemporary ability to accurately perform total Al analyses on previously ashed samples was confirmed with the relative standard deviation <1%.

### 2.4. Calculation of Target Hazard Quotient (THQ) and Estimated Daily Intake (EDI) of Al

The EDI was calculated with Equation (1), while the THQ was calculated with Equation (2) [12]. A THQ value being <1 indicates no significant risk of Al contamination to human health, whereas a THQ value being >1 indicates potential adverse effects to human health [24].
EDI = (C × E_F_ × E_D_ × F_IR_)/(W_AB_ × T_A_ × 1000)(1)
THQ = EDI/RfD(2)
where C is Al concentration in tea (μg/g); F_IR_ is average consumption of tea (8 g/person/day); E_D_ is exposure period (70 years); E_F_ is exposure frequency (365 d/year); T_A_ is duration of exposure (E_F_ × E_D_); and W_AB_ is average adult body weight (60 kg) [23]. According to the U.S. Environmental Protection Agency (USEPA), the RfD (Reference Dose) value of Al is 1.0 μg/g/day [25].

### 2.5. Statistical Analysis

The results of the chemical analysis were analyzed using one-way ANOVA regression analysis and Tukey’s test, which is available in the XLSTAT 2010 (Addinsoft, New York, NY, USA) and Microsoft Excel 2010 software packages (Microsoft Corporation, Redmond, WA, USA). The significance level was defined at *p*-value ≤ 0.05.

## 3. Results and Discussion

### 3.1. Al Content in Leaves at Different Positions (Leaf Ages) in Two Seasons

Figure 1 shows the Al concentrations in the composite tea leaves of four cultivars taken from specific positions on the stem in the winter and summer. The relationships between Al concentrations and positions of tea leaves of the four tea varieties in winter fit a bi-phase linear model much better than a monomial function (Figure 1 top panel), whereas those of the summer samples fit an exponential model well (Figure 1 bottom panel). The Al concentrations versus the top-5 leaf positions in the winter season followed one linear regression, while those versus leaf 6–11 fit another linear regression. The coefficients of determination (R^2^) were ≥ 0.968 (Figure 1, top panel). The slope values of the linear regression of the winter leaves 6–11 of the Yabukita, Yutaka Midori, and Ohiwase cultivars were 1.7–2.5-fold of the shoot + 2 leaves to 5, whereas that was 0.86-fold for the Mealani cultivar, which may be attributed to cultivar differences. In summer, the leaf positions and Al concentrations were exponentially correlated (R^2^ ≥ 0.900) (Figure 1, bottom panel). The results indicate that total Al content varies among cultivars, leaf positions, and seasons. The Al concentrations in tea leaves are probably related to leaf growth and age. Growth is at least a factor of dilution, which may be attributed to the bi-phase linearity (Figure 1 top panel). A linear model is user friendly for applications.

Figure 2 shows the average Al concentrations in the leaves from all of the cultivars at different positions on the tea plants grown in the winter and summer seasons. The Al concentrations in the winter leaves were higher than those in the summer leaves, although those were not significantly different between the two seasons at specific leaf positions. There is also a clear trend of higher Al concentrations in the lower (older) leaves. The data have confirmed the previous observations of increasing amounts of Al in “older” leaves compared with “younger” leaves [14,15] and further defined the Al content as a function of leaf positions. The average Al concentration reported in the “young” leaves at 381 μg/g [14] was similar to leaf 2 in winter and summer in this study (Figure 2). The Al concentration of 4200–7000 μg/g in the “old” leaves [14,15] was greater than that found in this study (Figure 2). It is noteworthy that the “young” and “old” leaves in the previous studies were not defined.

No previous work has shown the variations of Al concentrations in leaves at specific positions on the stem in combination with seasonal change. However, studies have been done on seasonal changes of polyphenol, catechin, antioxidants, and certain minerals in different cultivars of tea [26,27]. For example, shoots of Australian-grown *C. sinensis* showed higher levels of the flavanols epigallocatechin gallate, epicatechin gallate, and epigallocatechin during the warmer months compared with the cooler months [28]. A similar trend is observed with total phenolic content and antioxidant activity from *C. sinensis* shoots grown in Turkey [29]. In general, the concentrations of N, P, K, Ca, Mg, S, and Mn in tea leaves grown in Turkey were higher in the cooler months than those in the warmer months [27]. These results are probably attributed to slower growth in the cooler seasons. Much of the past research on mineral composition of tea leaves have been conducted in regions where all four seasons exist, and harvest is limited to spring and summer. However, seasons are less pronounced in Hawaii’s climate, where tea leaves can be harvested throughout the whole year. In this study, although the seasonal change had a small effect on the Al content of leaves at different positions of these four tea cultivars plants, the data may be useful to determine the optimal season, cultivars, and leaf position(s) for harvesting leaves to manage the total Al concentration in the tea product.

### 3.2. Comparison of Al Content in Leaves of Four Cultivars

Figure 3 shows a comparison of the Al concentrations among the four cultivars at each leaf position. In general, the variations in the Al content of the leaves among the four cultivars in summer were larger than those in winter. The differences in total Al in older leaves were larger than those in young leaves. This is similar to what was found in the first six leaf positions of the four cultivars of *C. sinensis* grown in Sichuan, China [30], although the average Al concentrations were approximately 6-fold higher than the data presented for Hawaii (Figure 2). When the tea leaves in fields are not identical and not maturing at the same rate is considered, the data variations seem quite plausible.

### 3.3. Potential Exposure to Al from Tea Consumption

Table 1 shows the comparison of Al content in food and tea products. The Al concentrations found in commercial tea products ranged from 123 to 3260 µg/g [6,9]. Al concentrations in the tea infusions were approximately 0.4–13.0 µg/mL [20]. In comparison, the Al concentrations found in commercial powdered tea ranged from 478 to 1229 µg/g (Table 1). The difference between powdered tea and tea infusions prepared with dried tea leaves is that the entire content is ingested when consuming powdered tea in any form, which allows for a much greater exposure of Al to the body compared with drinking an infusion in which the leaves are steeped into a liquid and not ingested.

As the Al RfD is 1 μg/g body weight per day, the THQ values equal EDI values. The EDI and THQ values of Al in the tea leaves of the four cultivars were all less than 1 (Table 2), indicating little threat to human health. The tolerable weekly intake (TWI) of Al intake was set at 1 μg/g body weight/week [8], and the tolerable intake of Al per day was 8.57 mg/day based on the average adult body weight (60 kg). When 2.0 g of Yutaka Midori (shoot + 2 leaves) were used to prepare an infusion in 250 mL of water, the Al concentration was 0.13 µg/mL (0.04% extraction efficiency) (Table 1). Using the Yutaka Midori cultivar as an example, the average 60 kg individual would need to ingest over 66 L/day of Hawaii tea for the sake of exceeding the safety limits. Similar results can be assumed for the other cultivars in the present study (Figure 3). It is quite possible to reach the TWI by consuming tea with other food products containing high Al concentrations (Table 1); however, it is important to understand that it is difficult to estimate the amount of Al that is available for daily absorption by the body. Research has suggested that the absorption of Al from tea is estimated to be less than 1%, yet it is still unknown how much Al is bound to other matter due to the varying levels of speciation of Al [31].

### 3.4. Al Content as a Measure of “Tea Leaf Quality” in Hawaii Teas

The concentrations of Al in Hawaii tea leaves harvested during the winter and summer seasons appeared to be similar among the four cultivars and showed a relatively constant increase with each successive leaf position. If a “high quality” tea product is often considered to be made of the shoot plus first two leaves, particularly for green tea, then the Al concentrations in a Hawaii tea may range from 150 to 200 µg/g (Figure 2). If the level is higher than 300 µg/g, then it becomes evident that the lower-level tea leaves might be used to make these products. However, the values in Figure 2 cannot be used for determining the leaf positions in teas produced from other countries. For example, the Al concentration from some tea leaves grown in Sichuan, China [30] appears to be about 6-fold higher than that from leaves grown in Hawaii at five leaf positions. It is noteworthy that the island of Hawaii (i.e., Big Island) has active volcanoes. An interest in other heavy metals in tea is, therefore, warranted in future studies, particularly in exposure to multiple toxic metals and risk assessments.

## 4. Conclusions

The present study was conducted to measure Al concentrations in leaves at different positions from four cultivars of the *C. sinensis* plant. The results agreed with the previous studies, wherein a higher Al concentration was seen with older leaves. Overall, there is a trend of increasing Al concentrations with lower (older) leaf positions. Seasonal variation was seen with higher levels of Al found during the winter months compared with the summer months. Individual cultivars showed similar trends of increasing Al concentrations; however, significant differences were shown with specific leaf positions for both seasons. This general pattern of higher Al concentrations during the winter months is most likely attributed to the slower growth in the winter. The calculated risk values for Al meant that there was negligible consumer health risk at the tea leaf positions of the four cultivars in Hawaii. Furthermore, the overall consistency of the results presented in this study, for both the winter and summer seasons, as well as among the cultivars, provided an opportunity to create a standard curve to estimate Al concentration, and therefore the “quality” evaluation in Hawaii-grown tea products. The results of this study may be useful for tea quality estimation and directing harvesting practices. Future studies may compare the Al concentration in leaves by geographical origin, soil acidity, cultural and processing techniques, and growing conditions and agronomic practices such as irrigation and shading.

## Figures and Tables

**Figure 1 ijerph-19-11952-f001:**
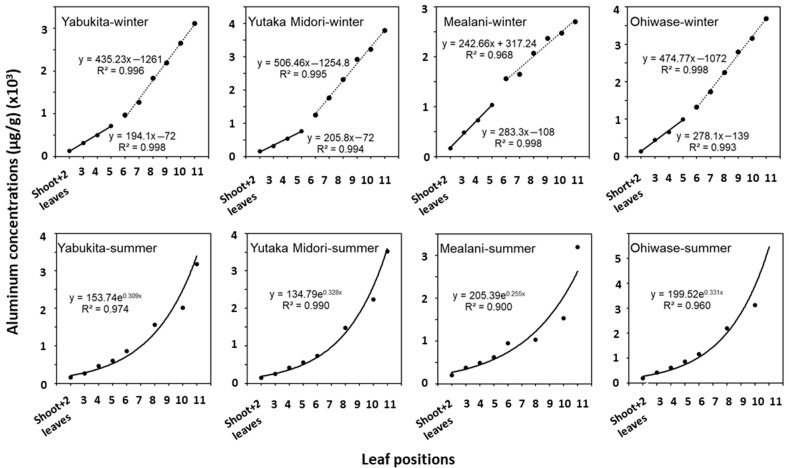
Relationship between the leaf positions and the mean concentrations of Al. Leaf positions 7 and 9 were not part of the original sampling scheme since they were known to be older than the leaves typically harvested (see Sample Collections).

**Figure 2 ijerph-19-11952-f002:**
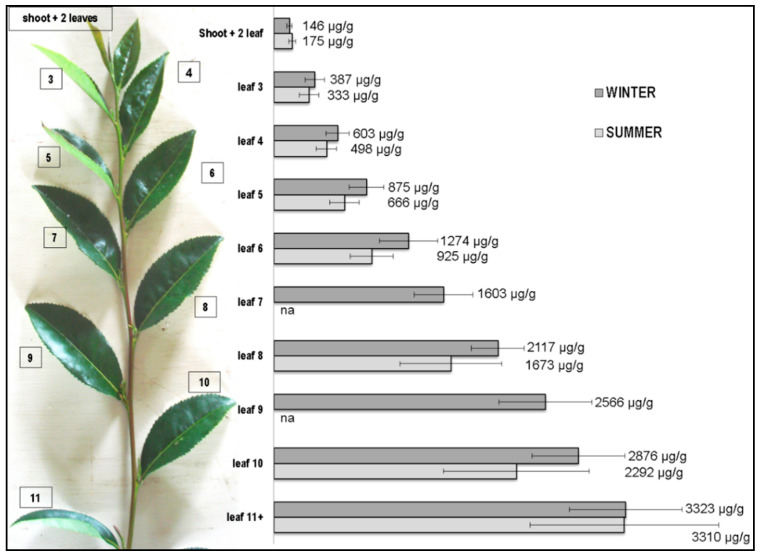
Seasonal comparison and correlation of average Al concentrations and tea leaf positions. Leaf positions 7 and 9 were not part of the original sampling scheme since they were known to be older than the leaves typically harvested (see Sample Collections).

**Figure 3 ijerph-19-11952-f003:**
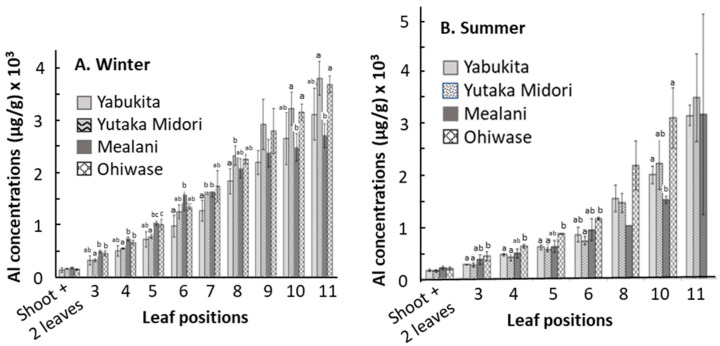
Comparison of Al concentration in winter (**A**) and summer (**B**) tea leaves among four cultivars at specific leaf positions. Different lower letters indicate statistically significant at *p* ≤ 0.05.

**Table 1 ijerph-19-11952-t001:** Comparison of Al Content in food products and *C. sinensis*.

	Al Concentration (μg/g)	Al (µg) per Serving
Food Item ^1^		
Processed American Cheese Slices	14–470	270–8900 per slice (19 g)
Pancake Mixes	19–1200	770–57,000 per 1/3 cup (40 g)
Baking Powder	18,000–28,000	72,000–112,000 per tsp. (4 g)
Non-dairy Creamer	110–590	260–1500 per individual packet (2 g)
*C. sinensis*		
Dry Tea Leaves ^2^	22.9–3260 ^3,4^	
Powdered Tea	478–1229 ^5^	1434–3687 per 240 mL ^6^
Yutaka Midori Infusion ^7^	0.13 μg/mL	31.2 per 240 mL
Prepared Tea Infusion ^8^	0.4–13 µg/mL	96–3120 per 240 mL

^1^ [11]. ^2^ Data from both unprocessed and processed tea leaves. ^3^ [18]. ^4^ [6]. ^5,7^ Data from the present study. ^6^ Values calculated from suggested preparation of traditional powdered tea (1 g/80 mL) [19]. ^8^ [20].

**Table 2 ijerph-19-11952-t002:** Estimated daily intakes (EDI) (μg/g body weight/day) of Al for consumers due to tea leaves.

Leaf Position	Estimated Daily Intakes (EDI) ^1^							
	Yabukita		Yutaka Midori		Mealani		Ohiwase	
	Winter	Summer	Winter	Summer	Winter	Summer	Winter	Summer
Shoot + 2 Leaves	1.69 × 10^−2^	2.08 × 10^−2^	2.04 × 10^−2^	2.00 × 10^−2^	2.21 × 10^−2^	2.71 × 10^−2^	1.85 × 10^−2^	2.53 × 10^−2^
Leaf 3	4.23 × 10^−2^	3.57 × 10^−2^	4.17 × 10^−2^	3.45 × 10^−2^	6.43 × 10^−2^	4.96 × 10^−2^	5.84 × 10^−2^	5.77 × 10^−2^
Leaf 4	6.59 × 10^−2^	6.19 × 10^−2^	7.21 × 10^−2^	5.53 × 10^−2^	9.64 × 10^−2^	6.52 × 10^−2^	8.72 × 10^−2^	8.33 × 10^−2^
Leaf 5	9.53 × 10^−2^	8.13 × 10^−2^	1.02 × 10^−1^	7.47 × 10^−2^	1.37 × 10^−1^	8.31 × 10^−2^	1.33 × 10^−1^	1.16 × 10^−1^
Leaf 6	1.29 × 10^−1^	1.14 × 10^−1^	1.66 × 10^−1^	9.76 × 10^−2^	2.08 × 10^−1^	1.27 × 10^−1^	1.76 × 10^−1^	1.55 × 10^−1^
Leaf 7	1.69 × 10^−1^	nc ^2^	2.36 × 10^−1^	nc ^2^	2.19 × 10^−1^	nc ^2^	2.31 × 10^−1^	nc ^2^
Leaf 8	2.44 × 10^−1^	2.08 × 10^−1^	3.09 × 10^−1^	1.97 × 10^−1^	2.76 × 10^−1^	1.37 × 10^−1^	3.00 × 10^−1^	2.94 × 10^−1^
Leaf 9	2.92 × 10^−1^	nc ^2^	3.89 × 10^−1^	nc ^2^	3.15 × 10^−1^	nc ^2^	3.72 × 10^−1^	nc ^2^
Leaf 10	3.54 × 10^−1^	2.69 × 10^−1^	4.30 × 10^−1^	2.98 × 10^−1^	3.30 × 10^−1^	2.04 × 10^−1^	4.21 × 10^−1^	4.17 × 10^−1^
Leaf 11	4.15 × 10^−1^	4.23 × 10^−1^	5.06 × 10^−1^	4.69 × 10^−1^	3.60 × 10^−1^	4.26 × 10^−1^	4.91 × 10^−1^	nc ^2^

^1^ Note: Since the Al RfD is 1 μg/g body weight per day, the EDI values are the same as THQ. ^2^ nc, not calculated.

## Data Availability

The data supporting the conclusion of this article will be available upon request from the corresponding authors.

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
