# Peer review of "Potential Health Risk of Aluminum in Four Camellia sinensis Cultivars and Its Content as a Function of Leaf Position"

_ijerph, 2022, doi:10.3390/ijerph191911952_

Round 1

Reviewer 1 Report

It is my recommendation to "reconsider after major revision" for reasons that include:

1. Nil mention of theory and research approaches on health-risk assessment of Al in foods in the Introduction that unnecessarily includes limited methodology and a questionable claim that Al is an essential element when at best (and in limited circumstances) it may be a beneficial element.

2. Inconsistent use throughout of the terms "concentration' and "content". This inconsistency needs to be resolved.

3. Very limited site description/s, including soil classification(s), age of the four tea cultivars, whether the tea plants were irrigated and/or fertilized and if so how and how often. Moreover, no clear reason is provided to account for incomplete samplings of leaf positions in Summer season. The shortfalls equate to 80% of expected samples for the first-two listed cultivars and 70% for the remaining two cultivars. These sampling shortfalls impact on winter-summer data comparisons. For example, grand median Al concentrations for all winter samplings of the four cultivars is 1441 mg Al/kg but decreases to 982 mg/kg when leaf positions common with those sampled in summer were compared. The equivalent grand median concentration for summer-sampled leaves is only 678.5 mg Al/kg.

4. Vague detail on the sampled leaves (e.g. were petioles included or removed, etc.) and on methodology for removing all surface contamination sufficient to ensure its full removal.

5. Also lacking are details on analytical measurement quality, noting some surprising differences within the data-sets for winter and summer seasons. For example, SD's expressed as percentages of mean concentrations for Winter and Summer samplings (cultivar Yabukita) were 21.3% and 9.7%, respectively. For Yutaka Midori, corresponding data were 7.1% and 15.8%; for Mealani the values were 8.9% and 23.9%; while for Ohiwase, corresponding values were 6.5% and 11.6%, respectively. The authors need to double check their data and (if confirmed) discuss whether the data variations may be due to sampling and /or analytical error or a reflection of crop health or similar.

6. The authors across Lines 40-41 quote a preferred growing-temperature range of 15—30oC, yet across Lines 78-79 of Materials and Methods claim (without comment) a favourable temperature range in Hawaii for tea production of 13  24oC. This is inconsistent.

7. The top row of Figure 2 has linear equations for variations in Al concentrations for different tea leaves with negative intercepts of from -85 to -636 mg Al/kg, which are impossible. The authors should select and present best-fit equations with more realistic intercepts.

8. No data presented to justify line 178—179 statement “The higher the leaf maturity is, the thicker the palisade tissue is, and the higher the Al content is.” And note here and in many other places there is overuse of “the”.

9. The graphical data in Figure 3 (A and B) is mostly a duplication of data in Table 1, although units are unnecessarily different: be consistent.

10. Manuscript contains both imperial and SI units. Recommend use of the latter.

11. Some 52 references seem more than needed for this relatively simple study. Moreover, the listing contains several discrepancies. For example, country names in at least references 18, 19, and 32 should be capitalised; book authors’ names in reference 25 should be capitalised; the journal name for reference 47 is lacking; while abbreviation for journal title in reference 46 requires double-checking.

Reviewer 2 Report

Well-written and straight-forward paper. Relatively much attention is paid to Al contents in tea leaves  (analytics) compared to the risk assessment part. The latter is carried out in a very basic deterministic way rather than stochastic one and may be some wording can be added to this.

page 7, line 246. It is mentioned that a THQ > 1 results in an obvious risk to human health. The word obvious is too strong in this context because of present safety margins in the followed approach. This should be changed in something like 'that potential adverse effects to human health may occur' 

page 8, line 258. No THQ-values are reported in Table 3 as stated! As the Al reference dose is 1 mg/kg bw day, the figures remains the same. Please add this.

The focus of the paper and risk assessment is on Al, a single compound. Is it possible that tea leaves may contain other inorganic/toxic elements and thus a combined exposure/risk assessment (HI; hazard index) should be  considered as well?

Reviewer 3 Report

This manuscript entitled “The Potential Health Risk of Aluminum in Four Camellia sinensis var. sinensis Cultivars and Its Content as a Function of Leaf Position”, which is an interesting research. In this study, leaves from four different cultivars of Camellia sinensis var. sinensis grown in Hawaii were analyzed for Al concentrations at 10 different leaf positions. Additionally, each cultivar was harvested in the winter and summer to determine seasonal variations of Al concentrations in the leaves. This manuscript has sound methods and statistical analyses, and well organized. And, the results of this paper may be useful for directing harvesting practices and to estimate the tea quality. However, I have some comments to enhance its quality as follows:

1.      Materials and methods: fresh leaf samples were collected from the tea plants grown at the University of Hawaii-College of Tropical Agriculture and Human Resources Mealani Research Station. As far as we know, the growth environment affects the accumulation of aluminum in tea plants. Can the aluminum content in the experimental samples represent the aluminum content in the actual production tea garden? If aluminum levels in tea grown in Hawaii were at safe levels, the significance of this study might not be as relevant.

2.      Materials and methods: Some details, for example, tea garden fertilization management, tea tree pruning management, etc should be add on.

3. Results and Discussion: Al Content in Leaves 7 and 9 in summer were not available? Please repeat the experiment to confirm whether the data is correct?

Round 2

Reviewer 1 Report

I still observe duplicated data, unequal sampling designs in winter & summer, & questionable analytical quality control.

Round 3

Reviewer 1 Report

Referee Comments on revised Manuscript submitted to Int. J of Environmental Research and Public Health: “The Potential Health Risk of Aluminum in Four Camellia sinensis var. sinensis Cultivars and Its Content as a Function of Leaf Position” by Huijuan Yang et al., August 2022

PREAMBLE

These comments pertain to a draft manuscript on the above topic, circulated mid-August 2022, following minor changes to an earlier draft. The study is reliant mostly on analytical results for tea leaf samples (including petioles) from four varieties of tea grown at a single location in Hawaii. Samplings in spring and summer, respectively, occurred in July 2009 and February 2010, which was 12 to 13 years ago. Analysis for Al occurred in the University of Hawaii’s Agricultural Diagnostic Service Centre, with no details on when the analyses occurred or on measurement-quality for total Al at that time. The authors need to explain the 12-to-13-year delay between data gathering and reporting in addition to supplying evidence from interlaboratory proficiency programs or similar that total Al results were both accurate and reliable. Uncertainty in measurement is important as is the quantum of data variability previously highlighted, such as SD's of 21.3% and 9.7% when expressed as percentages of mean concentrations for Winter and Summer samplings of cultivar Yabukita, respectively. Moreover, it is an experimental weakness that different leaf-sampling positions were utilised in winter and summer

Overall, there is a gross error, english expression requires improvement, while many unnecessary words (overuse of “the” is an example) and phrases can be removed without loss of technical content. The opening sentence of the Abstract is but one example. Additional feedback/comments follow under a series of headings. In addition, suggested words are offered for early sections / sub-sections of the manuscript, indicated by inverted commas.

ABSTRACT

Delete opening sentence and provide additional examples of summary results and conclusions drawn from the included data.

INTRODUCTION

The following text provides a suggested opening for the manuscript. It is offered as the present “opening”, which contains data on what quantities of tea leaves USA imports and consumes, is likely of minimum interest to most readers.

“It is well known that the tea plant accumulates Al in its leaves to a greater extent than most other edible plants, where it functions as a beneficial element (  ). Culturally, tea prefers soils with pH values ranging 4.5--5.5 in well-watered tropical environments, such as those across parts of Hawaii.  And since the daily allowable intake of Al is 6­14 mg/day for teens and adults (Joint FAO/WHO Expert Committee on Food Additives), it is appropriate to understand how concentrations in fresh tea leaves varies across countries and varieties and with season and place on the plant.”

“Information already exists on Al concentrations in tea leaves from Australia (), China (), India () Sri Lanka () and Turkey (), but was lacking for Hawaii, where there is increasing interest in the culture and commercial exploitation of multiple varieties of tea being evaluated in experimental plantations. Etc, etc, etc. Short-comings of existing research on Al in tea include a lack of understanding on effects of seasonal conditions and leaf positions on Al concentrations and how any differences might affect daily allowable intakes of Al in the end-product.”

Certainly give context by mentioning an entry pathway into humans of Al is via tea infusions through brewing and ingestion of tea, which adds to accidental inputs via fruits, vegetables, cereal products, etc. Confusion in the literature over descriptions / definitions of “young” and “old” leaves as well as the terms “shoots/bud plus two or three leaves” also exists and warrants clarification.  It would also be appropriate to mention points made around lines 425 and several beyond, specifically to indicate Al concentrations of from 123 to 3260 μg/g exist in commercial tea products [36,37].

Now complete the Introduction with something like “In this study, we examined Al concentrations from different leaf positions of four mature C. sinensis cultivars grown in Hawaii, analyzed these for total Al, assessed differences in Al among leaf positions, seasons and cultivars, then considered implications for human health and tea quality.”

MATERIALS AND METHODS

“Study site. Study site was at the University of Hawaii’s, College of Tropical Agriculture and Human Resources, Mealani Research Station, Honolulu. The tea-production field at Mealani is 610 m above sea level. Annual mean temperatures range from 13°C to 21°C. Soil is a Honokaa Series in the Andisol Order (Ref). Mean annual rainfall of 1651 mm was supplemented by timer-operated drip irrigation. The tea fields were planted from 1999 to 2004. Soil and tissue tests were used to guide fertilization, typically by monthly application of granular fertilizers supplemented by fertigation as needed. Soil pH to 15 cm typically ranges from X to Y (method).”

“Sample collections.  Fresh leaf samples from four selected cultivars of C. sinensis (Yabukita, Yutaka Midori, Mealani, and Ohiwase) at 10 leaf positions commencing at the stem-tip (shoot plus two leaves, leaf 2…leaf 10) were collected for subsequent analysis for total Al. The cultivars were chosen because they were of interest for distribution to commercial tea growers in the State of Hawaii, USA. Representative leaf samples of each cultivar were processed separately by cultivar and replication. Specifically for each leaf position, leaves including petioles were individually detached from the stem, wiped with a clean damp cloth to eliminate surface contaminants, then placed in a package labelled with cultivar name, leaf position, and replicate number for shipment to the Agricultural Diagnostic Service Center, University of Hawaii, Manoa. Moreover, leaf samples (with petioles) of each cultivar were collected in triplicate during summer (July 2009) and winter (February 2010) to determine seasonal variability. Leaf positions 7 and 9 were not part of the original sampling scheme since they were known to be older than leaves typically harvested. Those leaf positions were added in the subsequent winter-based sampling due to observed differences between adjacent leaf positions (see Figures 1 and 2). In addition, five different commercial, powdered tea samples were purchased in Hawaii for comparison with dry leaf and infusion data. The powdered samples were labelled as “matcha” and a product of Japan. {Can we assume these teas were sampled in accord with guidelines in ISO 1839:1980 (en) Tea — Sampling?} All leaf and powdered tea samples were kept in dark, moisture-proof containers at room temperature until analyzed.”

“Shoot-plus-two leaf samples of Yutaka Midori cultivar were used for infusion preparations. These were prepared by steeping 2.0 g portions in 200 mL deionized boiling water (18 mΩ) for 30 min in a covered container. Resulting infusions were filtered through Whatman no. 42 paper, then diluted to 250 mL in volumetric flasks and allowed to cool to room temperature prior to analysis. Infusion samples were prepared in triplicate and stored in covered plastic vials at 3-5°C until analyzed. It was assumed no precipitations occurred within the stored infusions.”

“Chemical analysis. The samples of tea leaves were further cleaned with damp paper towels to remove visible dust, placed in a forced-draft oven at 55°C for 12 h, then ground with a Wiley mill to 2 mm. A 0.50 g sample was dry-ashed in a porcelain crucible for 4-6 h at 500 °C in a muffle furnace, cooled, dissolved in 1 mol/L HNO3, evaporated to dryness, then ashed again for 1 h. The resulting residue was dissolved in 25 mL of 1 mol/L HCl [23]. All prepared samples (dried leaf, powdered tea and infusions) were then analyzed for total Al by ICP-OES, using a wavelength of ??, according to the AOAC Official Method 953.01 [24,25]. The laboratory’s contemporary ability to accurately perform total Al analyses on previously ashed samples was confirmed by ???????”

“Statistical analysis. The results of the chemical analysis were analyzed using one-way ANOVA, regression analyses, and Tukey’s test, available in the XLSTAT 2010 (Addinsoft, New York, NY) and Microsoft Excel 2010 software packages (Microsoft Corporation, Redmond, WA). Significance level was defined at p-value ≤ 0.05.”

Referee’s Comment. The above text is based on what could be deduced by the referee from Methods and Material descriptions. That said, as there are no “leaf 11” data in regressions shown in Figure 1, text on the second line of ‘Sample collections’ has been changed to: ‘shoot plus two leaves, leaf 2…leaf 10’.

RESULTS AND DISCUSSION

Referee’s Comments.

·      Excessive duplication of data is apparent, inclusive of three multi-graphical Figures (Figures 1, 2 and 3). Moreover, the regressions for Al in winter-season cultivars are highly suspect as it seems data for all four cultivars were arbitrarily split into two populations prior to the fitting of linear models. Indeed, non-linear models are available that include the inflection point as a parameter of the model. However, the data trends for all four cultivars can be explained with high confidence (average r2 values of 0.99) with continuous, power function models. Relevant equations for the four cultivars (in winter) are presented in the following table.

Cultivar (winter)

Power function

r squared

Yabukita

y = 37.55 x1.8438

0.9976

Yutaka Midori

y = 37.601 x1.9475

0.9908

Mealani

y = 73.921 x1.5804

0.9584

Ohiwase

y = 47.882 x1.8443

0.9955

·      Importantly, the authors text relating to Figure 1 on lines 206-207 and 211 incorrectly refer to R squared values as ‘correlation coefficients’. The correct term is ‘Coefficient of determination’, noting this is the square of the corresponding correlation coefficient (r). Also, why only 7 data-points in the summer graphic for Ohiwase?

·      It is unusual to refer to Figure 2 in the manuscript-text prior to mention of Figure 1. Indeed, the Figure 2 title is lacking in detail regarding the data included. Also is the leaf and stem picture in Figure 2 original to the authors? The authors need to do more with the aggregate data presented or delete Figure 2 entirely, as it adds little that is not already known.

·      In this reviewer’s, Figure 3 is perhaps the most informative. Little would be lost if both Figures 1 and 2 were deleted entirely.

·      Regarding the sub-section on ‘Comparisons of Al in tea leaves’, it is far to wordy to justify its full retention. As a preferred alternative, the authors could summarise salient points through to line 410 in a tabulation by cultivars and seasons.

·      The sub-section on ‘Potential Exposure to Al from Tea’ is introductory in part, often with unnecessary detail. At most, only include data on Al in tea in Table 1 as further information on Al contents of food products is readily available. Perhaps all of what is retained could be incorporated into a sub-section on ‘Al content and quality of Hawaiian Teas’.

ABBREVIATIONS

The revised manuscript is inconsistent throughout on abbreviations, including for aluminum. Indeed, there are several examples of inter-use of Al and aluminum. And the same applies to other terms. Suggest if defined in this section, the same term should not be redefined elsewhere in the text where only the symbol or abbreviation should be used.

REFERENCES

44 listed references is too many for this relatively simple study.

UNITS

Manuscript contains both imperial and SI units. Recommend use of the latter.

END
